# Autophagy in Synucleinopathy: The Overwhelmed and Defective Machinery

**DOI:** 10.3390/cells8060565

**Published:** 2019-06-09

**Authors:** Marie-Laure Arotcarena, Margaux Teil, Benjamin Dehay

**Affiliations:** 1Univ. de Bordeaux, Institut des Maladies Neurodégénératives, UMR 5293, F-33000 Bordeaux, France; marie-laure.arotcarena@u-bordeaux.fr (M.-L.A.); margaux.teil@u-bordeaux.fr (M.T.); 2CNRS, Institut des Maladies Neurodégénératives, UMR 5293, F-33000 Bordeaux, France

**Keywords:** Parkinson’s disease, multiple system atrophy, synucleinopathy, α-synuclein, macroautophagy, chaperone-mediated autophagy, mitophagy, neurodegeneration

## Abstract

Alpha-synuclein positive-intracytoplasmic inclusions are the common denominators of the synucleinopathies present as Lewy bodies in Parkinson’s disease, dementia with Lewy bodies, or glial cytoplasmic inclusions in multiple system atrophy. These neurodegenerative diseases also exhibit cellular dyshomeostasis, such as autophagy impairment. Several decades of research have questioned the potential link between the autophagy machinery and alpha-synuclein protein toxicity in synucleinopathy and neurodegenerative processes. Here, we aimed to discuss the active participation of autophagy impairment in alpha-synuclein accumulation and propagation, as well as alpha-synuclein-independent neurodegenerative processes in the field of synucleinopathy. Therapeutic approaches targeting the restoration of autophagy have started to emerge as relevant strategies to reverse pathological features in synucleinopathies.

## 1. Introduction

Synucleinopathies are neurodegenerative diseases characterized by the presence of alpha-synuclein (α-syn)-positive intracytoplasmic inclusions in the central nervous system (CNS), such as Parkinson’s disease (PD), multiple system atrophy (MSA), and dementia with Lewy bodies (DLB). PD is the most prevalent synucleinopathy worldwide, with 1–5% of the population over the age of 60 developing this disease [1]. PD is clinically characterized by a triad of motor symptoms (resting tremor, rigidity, and akinesia) caused by dopaminergic striatal depletion due to the loss of dopaminergic neurons in the substantia nigra (SN). The major neuropathological hallmark of PD is the presence of α-syn-positive intracytoplasmic inclusions called Lewy Bodies (LB), which accumulate in neurons [2,3]. DLB is the second most prevalent synucleinopathy, clinically characterized by parkinsonian syndrome associated with cognitive deficits. DLB patients exhibit neurodegeneration in the SN, as well as LB deposition in the brain, but DLB differs from PD by the presence of amyloid plaques in cortical areas [2]. MSA is a rarer synucleinopathy, affecting 1–9/100,000 persons worldwide, divided into two clinical phenotypes: the MSA-parkinsonian with levodopa non-responsiveness parkinsonian syndrome due to neurodegeneration in the nigro-striatal pathway, and the MSA-cerebellar with gait, speech, and limb ataxia, as well as oculomotor dysfunction, due to neurodegeneration in the olivo-pontocerebellar system. As part of synucleinopathy, MSA patients’ brains exhibit α-syn-positive intracytoplasmic inclusions, which accumulate mainly in the oligodendrocytes, called glial cytoplasmic inclusions (GCI) [2].

Although synucleinopathy regroups heterogenous clinical disorders, the neuropathological common denominator is the accumulation of the 140-amino acid protein, α-syn, in different brain regions of PD, DLB, and MSA patients. The α-Syn accumulation leads to the formation of misfolded α-syn species that aggregate in the cytoplasm, forming LB and GCI. Although the trigger phenomenon remains to be elucidated, three cellular factors have been suggested to play a role in α-syn aggregation: (i) increased expression of the *SNCA* gene encoding α-syn, (ii) post-translational modifications or mutations favoring α-syn aggregation, and (iii) impaired α-syn protein degradation. The α-Syn degradation is ensured by both the ubiquitin-proteasome system [4] and by the autophagy-lysosomal pathways (ALP) [5] protein degradation machinery. The signals responsible for targeting α-syn to either degradation pathway remain not fully understood. 

ALP was shown to be implicated in the degradation of monomeric, small, soluble oligomeric species, as well as aggregated forms of the α-syn protein. ALP is a cellular proteolytic system which allows the degradation of long-lived proteins, protein aggregates, and abnormal organelles through both macroautophagy (MA) and chaperone-mediated autophagy (CMA) processes [6]. MA degrades cellular waste after the fusion of the autophagosomes carrying the material with the lysosome containing the enzymatic material [6]. CMA is a selective pathway that degrades proteins after recognition of pentapeptide (KFERQ-like motif) sequence by the cytosolic chaperone heat-shock cognate 70kDa protein (Hsc70) and delivery to the lysosome [6]. Here, we review clinical and experimental evidence focusing on PD and MSA, suggesting that defects in ALP machinery actively participate in α-syn accumulation in synucleinopathies and neurodegenerative pathological process.

## 2. Clinical Evidence of Autophagy Implication in Synucleinopathies

### 2.1. Genetic Evidence of Autophagic Involvement in Familial PD Cases

Genetic studies have been conducted for PD identifying familial forms of the disease representing approximately10% of the PD cases with hereditary parkinsonism and earlier onset of the disease. 

Among PD-linked genes, the first autosomal dominant mutation identified for PD was localized in the *PARK1/SNCA* gene encoding for the α-syn protein. This point mutation constitutes the A53T substitution in the *SNCA* gene, identified in Italian and Greek families [7]. In 1998, Kruger and collaborators identified the A30P point mutation in a German family [8] and the E46K point mutation was described in a Spanish family in 2004 by Zarranz and colleagues [9]. All these three *SNCA* point mutations were associated with α-syn accumulation and PD development, and were suggested to participate in autophagy impairment [10,11,12]. The autosomal dominant mutations in the *PARK8* gene encoding the leucine rich repeat kinase 2 protein (LRRK2) were reported to be the most known genetic cause of familial PD cases [13]. As LRRK2 is composed of multi-domain proteins, mutations in *PARK8* gene induce neurodegeneration by alteration in multiple cellular pathways, including the autophagy machinery [14]. The G2019S point mutation is the most common *PARK8* mutation, accounting for 5% of familial cases worldwide, and up to 42% of familial cases from North African Arab patients [15]. This mutation affected the kinase domain of the protein leading to impaired autophagy degradation. In 2006, two autosomal recessive mutations in the *PARK9* gene encoding for a lysosomal ATPase cation transporter (ATP13A2) in a Chilean family with early-onset Parkinson disorder were reported [16]. The ATP13A2 protein physiologically colocalizes with the lysosomal-associated membrane protein 2 (LAMP-2). The PD heterozygous *PARK9* mutations lead to an unstable truncated ATP13A2 protein, which is abnormally retained at the endoplasmic reticulum (ER) before its degradation [17]. Regarding genes involved in the CMA process, an autosomal recessive mutation in the *DNAJC6* gene, encoding DnaJ heat shock protein (Hsp) family member C6, was initially found in two members of a family in Palestine and was associated with fast-progressing parkinsonian syndrome [18]. Subsequent follow-up studies reported other *DNAJC6* mutations in families with very early-onset parkinsonism [18]. More recently, several groups also reported heterozygous mutations in the DnaJ heat shock protein family member C13 (*DNAJC13*) gene leading to pathological features resembling idiopathic PD [19]. Related to mitophagy-linked genes, *PARK2* gene encoding the E3-ubiquitin ligase protein Parkin was the second identified PD-linked gene with a large spectrum of autosomal recessive mutations observed in different familial cases [20,21,22,23,24,25]. Most of the *PARK2* mutations are deletions affecting the ubiquitin-like domain of the protein, disturbing the stabilization of other cellular proteins. The second mitophagy-linked gene constituted by *PARK6* gene mutations encoded the phosphatase and tensin homolog-induced putative kinase 1 (PINK1) protein is found in 1–9% of familial PD cases [26,27,28]. The majority of *PARK6* mutations affected the kinase domain of the protein, leading to a loss-of-function of the protein, physiologically involved in eliminating damaged mitochondria. Finally, autosomal recessive mutations in *PARK7* gene encoding the DJ-1 protein are rarer and lead to its reduced antioxidant activity [29,30]. 

Besides PD-linked genes, genome-wide association studies pointed to some polymorphisms in genes that are associated with increased risk of developing PD. Among them, the *GBA* gene encoding the lysosomal enzyme β-glucocerebrosidase (GCase) and involved in glycolipid metabolism is a well-validated PD-associated risk factor [31]. *GBA* mutations cause an inherited autosomal recessive lysosomal storage disorder, known as Gaucher disease (GD), characterized by a loss-of function of the GCase protein leading to accumulation of glucocerebroside in different organs, such as the liver, the blood, the spleen, the lungs, and the nervous system. Interestingly, 8–14% of PD patients carried mutations in their genome, suggesting an increased risk of developing PD [32]. Similarly, patients with GD carrying heterozygous *GBA* mutations exhibited higher risks of developing PD [33]. Strikingly, a recent study discovered a significant burden of new susceptible loci, likely damaging lysosomal storage disorder gene variants in association with PD risk, reinforcing the importance of lysosomal mechanisms in PD pathogenesis [34]. 

As no genetic forms or polymorphisms have been reported in the literature so far for MSA, no genetic evidence regarding this other synucleinopathy can be reviewed here. Nevertheless, genetic studies on PD already indicate the potential role of autophagic processes into PD pathogenesis.

### 2.2. Post-Mortem Evidence of Autophagic Involvement in Synucleinopathies

Post-mortem studies have provided numerous pieces of evidence of autophagy impairment in pathologically affected areas of the brains of MSA and PD patients compared to age-matched controls. In 1997, Anglade and collaborators already observed an impairment in the autophagic process, noticing a drastic decrease of autophagic vacuoles in the nigral dopaminergic cells on PD post-mortem brains [35]. The most striking example highlighting an ALP impairment associated with α-syn accumulation in synucleinopathies is the presence of the microtubule-associated protein 1A/1B-light chain 3 (LC3), an autophagosomal marker, within the LB of PD brains and within the GCI of MSA brains. In 2010, different groups showed that LC3 colocalized with LB and Lewy Neurites (LN) in the SN of PD patient brains [36], with an estimation of up to 40% of cortical LB [37] and up to 80% of nigral LB [38] being positive for LC3 in PD patients. Post-mortem studies on DLB patients—another synucleinopathy showing α-syn positive inclusions in cortical and cognitive areas—further corroborated this data, demonstrating co-immunostaining of LC3 with LB in the hippocampus [39] or in the temporal cortex [40]. Co-staining of LC3 and phosphorylated forms of α-syn in GCI were also observed in the oligodendrocytes of MSA brain patients [41], with up to 84% of colocalization found in pons tissue [42]. Taken together, these results suggest the accumulation and defective clearance of autophagosomes in synucleinopathies. In PD cases, Dehay and coworkers also observed that the transmembrane lysosomal transporter ATP13A2 is a component of LB in the SN of PD cases, as evidenced by 90% of ATP13A2 immunoreactivity in LB-positive dopaminergic neurons of the SN [43]. The autophagic regulator protein Beclin-1 was shown to be caspase-cleaved and increased in PD brains [44]. Finally, a more systemic autophagy impairment occurs in synucleinopathy, as both MA and CMA activity are decreased in cultured peripheral blood mononuclear cells of PD patients [45].

At a cellular level, alterations in the ALP could be explained by a decreased expression of the ALP transcription factor LIM homeobox transcription factor 1 beta (LMX1B) observed in melanized dopaminergic neurons of post-mortem PD brains [46]. Among lysosomal components, the amount of the lysosomal-associated membrane protein 1 (LAMP-1) [47] and the ATP13A2 protein [43,48] are reduced in PD patients in comparison with age-matched controls, suggesting a defect in the MA process in PD pathogenesis. Corroborating this data, Dehay and colleagues described an impairment in the lysosomal-mediated clearance of autophagic vacuoles with an accumulation of autolysosomes containing non-degraded material in fibroblasts derived from PD patients harboring ATP13A2 loss-of-function mutations [43]. Pontocerebellar fibers of MSA patients showed increased staining of LAMP-1 and LAMP-2 with abnormal random distribution into the cytoplasm compared to control tissues, suggesting an MA impairment that is also involved in MSA pathogenesis [49]. Tanji and colleagues further demonstrated a decreased expression of the *GABARAP* gene encoding for the Gamma-aminotubyric acid receptor-associated protein and the *GABARAPL2* gene encoding for the Golgi-associated ATPase enhancer of 16KDa (GATE-16), which both play important roles in autophagosome formation associated with impaired maturation of the proteins in the cerebellum of MSA patients [41]. Finally, Compagnoni and colleagues demonstrated an autophagic activation with LC3-II accumulation in the cytoplasm of iPSCs derived from MSA patient skin fibroblasts associated with defect in lysosomal clearance [50], confirming impairment on the MA pathway.

Regarding the CMA pathway, Alvarez and colleagues demonstrated decreased protein levels only for the isoform type A of LAMP-2 (LAMP-2A) rate-limited protein for CMA process in the SN and the amygdala of PD patients, which is associated with decreased Hsp70 protein levels [36]. Corroborating this data, Murphy et al. (2015) showed that decreased levels of LAMP-2A and Hsc70 were associated with α-syn accumulation at the early stages of PD [51], and α-Syn protein was also shown to be colocalized with Hsc70 in the oligodendrocytes from MSA brains [52], confirming an impairment in the CMA machinery.

Finally, alterations in lysosomal enzymatic activities, involved in both MA and CMA processes, have been also described in post-mortem brains of patients suffering from synucleinopathies. The lysosomal GCase enzyme activity was shown to be decreased in the cerebellum, the amygdala, the putamen, and the SN of PD brains from *GBA* mutation carriers [53]. Regarding sporadic cases, its activity is reduced in the cerebellum and in the SN by 24% and 33%, respectively [53], and in the caudate nucleus [54]. Its decreased activity was found to correlate with an increased α-syn amount at the early stages of the disease [55]. GCase activity was also demonstrated to be decreased in the cerebrospinal fluid (CSF) of PD patients compared to controls [56,57], as well as a decrease in α-amminosidase [50,56] and α-fucosidase [58] activity and increase in β-hexosaminidase [57] and β-galacosidase [58] activity. Reduced activity of lysosomal hydrolase Cathepsin D (CathD) was observed in the SN of PD patients [47] or in ATP13A2 mutant fibroblasts derived from PD patients associated with defective processing of cathepsins [43]. In MSA cases, alterations in hydrolases were also observed with increased levels of CathD found in the pontocerebellar fibers [49] and in the white matter of the cerebellum [41].

In conclusion, post-mortem analysis of PD and MSA patients pointed out the ALP, macroautophagy, as well as CMA as be part of α-syn accumulation and neurodegenerative processes in synucleinopathies.

## 3. Dual-loop Between Autophagy and α-Syn Pathogenesis

Multiple pieces of evidence suggested that defective ALP processes could participate in α-syn accumulation, aggregation, and propagation into neuronal cells, thus confirming the role of deficient autophagy machinery in synucleinopathy.

### 3.1. ALP Processes Participate in α-Syn Accumulation and Aggregation

As suggested by genetic and post-mortem studies, ALP machinery is impaired in synucleinopathies and seems to be involved in α-syn accumulation. Of interest, multiple studies have provided evidence that ALP participates in α-syn degradation, in addition to the ubiquitin proteasome system. 

In an α-syn transgenic mouse model, Klucken and collaborators observed the colocalization of α-syn positive inclusions with autophagic markers, such as LAMP-2 and LC3, as observed in patients’ brains [59]. In fibroblasts derived from PD patients harboring a loss-of-function mutation in the *PARK9* gene, an accumulation of α-syn protein was observed in parallel with the accumulation of lysosomes and decreased proteolytic activity [43,60]. Knocking-down ATP13A2 expression in primary cortical neurons also enhanced accumulation of endogenous α-syn protein in addition to lysosomal degradation impairment, which can be reversed in part by the expression of wild-type (WT) ATP13A2 in ATP13A2 knockdown cells [60], confirming that α-syn is degraded by the lysosomes. Tsunemi and collaborators showed that *PARK9* deficiency leads to defective lysosomal exocytosis through impaired lysosomal Ca^2+^ levels contributing to intracellular α-syn accumulation [61]. This study suggested that ATP13A2 regulated neuronal lysosomal exocytosis, which modulates α-syn intracellular levels [61]. Recently, Sato and collaborators generated a Tyrosine-Hydroxylase cell-specific Atg7 conditional knock-out (KO) mouse model that presented α-syn accumulation and LB-like inclusions (positive for α-syn, p62, and ubiquitin) after autophagy impairment [62]. Thus, α-syn is degraded by ALP processes and several studies aimed to determine the contribution of MA and CMA pathways in α-syn degradation. 

In PC12 cells that overexpress WT or mutated forms of α-syn (A30P, A53T), Webb and colleagues demonstrated that the induction of MA using the mTOR inhibitor rapamycin leads to an increased clearance of both WT and mutated forms of α-syn [63]. Conversely, increased levels of intracellular α-syn were observed after inhibition of the autophagosomal formation using the PI3K inhibitor 3-methyladenine (3-MA) or inhibition of the fusion of the autophagosomes with the lysosomes with BafilomycinA1 (Baf) [63,64]. In neuronal cells overexpressing WT α-syn, de novo α-syn aggregates colocalize with lysosomal markers CathD and LC3 [65]. Overexpression of Beclin-1 in this cellular model and in a transgenic α-syn mouse model leads to a decrease of α-syn aggregates but not monomeric forms, confirming an autophagic role in α-syn aggregate clearance [64,65]. On the contrary, Beclin-1 or Atg5 silencing in A53T-α-syn expressing neuroblastoma M17 cells induced a significant increase of α-syn oligomers and aggregates [66]. Altogether, these results suggest that the MA system is involved in clearance of α-syn oligomers or aggregates. 

Interestingly, another study showed that clearance of human α-syn-aggregates induced by rotenone exposure to COS-7 cells is decreased after lysosomal inhibition using Baf treatment, but not after macroautophagic inhibition using 3-MA treatment [64]. Likewise, Cuervo and colleagues also demonstrated that rat ventral midbrain dopaminergic neurons exposed to 3-MA did not inhibit α-syn degradation, whereas a significant increase of α-syn half-life was observed after exposure to the lysosomal proteolysis inhibitor ammonium chloride [67]. These observations suggested that even if lysosomes are clearly involved in α-syn degradation, a non-macroautophagic pathway also took part in α-syn aggregates clearance [64]. Interestingly, the α-syn protein sequence contains the CMA-recognition motif VKKDQ [68]. Mutations in the CMA recognition motif of the α-syn protein sequence dramatically decreased its translocation to the lysosomal lumen [67]. In the same line of evidence, knocking-down the rate-limited CMA receptor LAMP-2A in rat embryonic cortical neurons increased detergent-soluble and -insoluble α-syn levels [69]. In vivo LAMP-2A silencing in dopaminergic cells in a rat model also induced autophagic defects with accumulation of autophagic vacuoles surrounded by α-syn punctates, which were positive for ubiquitin [70]. Altogether, this emphasized that CMA could be the non-macroautophagic pathway contributing also to α-syn degradation.

Additionally, the expression of micro RNA (miRNA) targeting LAMP-2A or Hsc70 CMA proteins in SHSY5Y cells also led to α-syn accumulation after defects in CMA [71,72]. Interestingly, these miRNAs, such as miR106a, miR320a, and miR301 (for Hsc70), or miR224 and miR373 (for LAMP-2A) levels are upregulated in the SN of PD patients, suggesting a CMA-mediated α-syn clearance defect in pathological conditions that could be involved in their accumulation and aggregation. Further confirming these results, addition of the chaperone protein DJ-1 to the α-syn solution inhibited the production of α-syn protofibrils, whereas the addition of L166P parkinsonism mutated DJ-1 failed to inhibit the generation of α-syn protofibrils [73]. Knocking-out DJ-1 chaperone protein in SHSY5Y cells or in the parkinsonian MPTP mouse model induced increased levels of α-syn oligomers associated with decreased Hsc70 levels [74]. This suggested that DJ-1 deficiency enhanced the accumulation and aggregation of α-syn through a decrease of CMA-mediated degradation. Regarding other PD-linked proteins, loss-of-function mutations in the *GBA1* gene increased α-syn oligomeric accumulation, as demonstrated in vitro [75,76,77]. Likewise, decreased GCase activity after β-epoxide treatment in vitro [78] or in a GD rodent model [76,77,79] enhanced pathological α-syn accumulation and aggregation. Altogether, these data further highlighted the contribution of CMA impairment in pathological accumulation of α-syn.

A30P and A53T mutated forms of α-syn seemed to be specifically degraded by the MA process, as they were poorly internalized to the lysosomal lumen despite their high affinity to the LAMP-2A receptor [67]. This induced a compensatory toxic gain-of-function activation of the MA process to counteract the CMA blockade induced by these mutated forms [67,69], which was not sufficient to maintain α-syn at physiological levels. 

Altogether, these data suggest that α-syn is a macroautophagic and a CMA substrate (Figure 1a). Impairment of ALP in pathological conditions participates in α-syn accumulation and aggregation, as observed in synucleinopathies (Figure 1b).

### 3.2. ALP Pathways Participate in α-Syn Propagation

The prion-like hypothesis of α-syn in synucleinopathies suggests that it is able to propagate by cell-to-cell transfer into the CNS, as described in graft post-mortem studies [80,81,82,83] or in different experimental models [81,84,85]. Among the possible routes of propagation, membrane-bound vesicles of endocytic origin called exosomes were suggested to carry α-syn from one donor cell toward a recipient one [86]. Interestingly, α-syn protein was found to be increasingly associated with vesicle fractions from human CSF of LBD and PD patients compared to control subjects [87]. Isolation of exosomes from WT α-syn overexpressing SHSY5Y conditioned medium confirmed this association between α-syn and exosomes [88]. Incubation of naïve SHSY5Y cells to these α-syn-positive-exosomes showed the transmission of α-syn to the recipient cells [88,89]. The integrity of the exosomal compartment is essential for α-syn transmission, as its membrane disruption prevents the α-syn transmission to the donor cells [88,89]. Several studies confirmed this exosomal-mediated cell-to-cell α-syn transmission in vitro using different sets of co-culture techniques [89,90].

Interestingly, inhibition of autophagy using chemical treatments or Atg7-KO cells led to a significantly increased α-syn release in the conditioned media, and particularly in the exosomal fractions associated with increased levels of α-syn in recipient cells [88,89,90]. In contrast, activation of autophagy using rapamycin induced significant lower levels of α-syn exosomal release [88,89,90] and cell-to-cell transfer of α-syn in vitro. Systemic treatment of α-syn transgenic mice with Baf increased α-syn levels in the CSF, as well as neuronal death and neuroinflammation [91]. Likewise, injection of CSF-derived exosomes from LBD patients into mice brains increased the occurrence of α-syn-positive inclusion bodies in neurons, suggesting that the delivery of exosomal α-syn induces synucleinopathy [87]. These data suggest that autophagy impairment caused increased α-syn secretion and enhanced cell-to-cell transmission of α-syn via release of non-degraded α-syn-containing exosomes. 

Further confirming this hypothesis, Tsunemi and collaborators demonstrated that fibroblasts from patients carrying *PARK9* mutation produced higher amounts of multivesicular endosomes with sparse intraluminal vesicles, which represent exosome precursors, suggesting that the ATP13A2 protein may control the exosomal process [92]. Using ATP13A2 overexpression in neuroblastoma H4 cells and primary neuronal cultures, they showed that α-syn levels are decreased in association with a higher production of exosomes, confirming that ATP13A2 plays a critical role in α-syn transport mediated by exosomes [92]. Kong and collaborators suggested that ATP13A2 mediates exosomal α-syn transmission by controlling zinc homeostasis in these vesicles [93]. Finally, *GBA* KO SHSY5Y cells accumulated intracellular α-syn and increased α-syn extracellular release, which was taken up by recipient naïve cells [94]. This phenomenon was reversed by overexpression of WT *GBA* in KO cells. Transplantation of *GBA* KO cells into the hippocampus of human α-syn transgenic mice showed that grafted cells were more sensitive to host-derived α-syn transfer [94]. Thus, reduced GCase activity by *GBA* mutation or in pathological conditions enhances the cell-to-cell transmission of α-syn.

Altogether, these studies demonstrated that autophagic impairment present in synucleinopathies could enhance α-syn cell-to-cell transmission by increasing the amount of non-degraded delivered exosomes to the extracellular media, thus aggravating the α-syn spreading into the CNS (Figure 1b).

### 3.3. α-Syn Induces Alterations on ALP Processes

Several studies also reported a detrimental role of the protein α-syn on ALP, further contributing to the pathology through an endless cycle of cellular alterations. PC12 cells overexpressing WT or mutated A53T α-syn protein showed abnormal morphology with accumulation of aberrant vesicular structures associated with diminished lysosomal activity, suggesting an impairment of the autophagy machinery in correlation with α-syn levels [95]. 

WT α-syn overexpression induced autophagy inhibition through impairment in autophagosome formation [96,97,98]. One possible mechanism relates to a decrease in α-syn-induced Rab1 activity, which, in turn, perturbs the localization of the Atg9 protein involved in autophagosome formation at the early stage of the biogenesis [97]. Song and collaborators suggested that α-syn-mediated autophagy inhibition was caused by the interaction between α-syn and the autophagic stimuli trigger HMGB1, associated with decreased Beclin-1 protein levels. This prevented the physiological cytosolic interaction between Beclin-1 and HMGB1 essential to activate autophagy [96]. Moreover, dopamine-modified α-syn showed inhibitory effects on the CMA activity by: (i) binding the LAMP-2A protein at the lysosomal membrane, (ii) forming oligomeric complexes at the lysosomal membrane, and (iii) being poorly translocated to the lysosomal lumen, thus blocking the degradation of other CMA cellular substrates [98,99]. Finally, when α-syn transfers from cell to cell, autophagic impairment was also triggered in the recipient cell with an accumulation of enlarged deficient lysosomes, further contributing to the expansion of the pathology [100].

In parallel to WT α-syn effects on autophagy, pathologically mutant forms of α-syn induce alteration in ALP processes, per se. Overexpression of mutated A53T α-syn decreases the autophagy activity in PC12 cells. A53T α-syn lacking the CMA recognition motif expressed in the same cells did not alter autophagy activity, suggesting that the mutant A53T α-syn induced alterations only in the CMA-mediated degradation process [98]. In this situation, the MA pathway was upregulated to compensate for the CMA deficit but resulted in detrimental conditions that were partly responsible for the observed cell death. Disruption of CMA-mediated degradation induced by A53T α-syn also led to abnormal accumulation of the myocyte enhancer factor 2 (MEF2D) in the cytosol. This compromised its binding to DNA and its consequent role in gene regulation of multiple cellular functions, inducing cellular dyshomeostasis [101]. The mutant E46K α-syn induced impaired autophagy through inhibition of autophagosome formation in an mTOR-independent pathway [102]. It was suggested that E46K α-syn reduced the phosphorylation of Bcl2 protein through upstream JNK1 inactivation, provoking the sequestration and the loss-of-function of Beclin-1 in the Bcl2/Beclin-1 complex [102]. The mutant A30P α-syn was suggested to inhibit autophagy by increasing the translocation of the repressive autophagic transcription factor ZKSCAN3 to the nucleus in a JNK-dependent manner [11].

In α-syn preformed fibrils-treated HEK293T cells, phosphorylated α-syn (P-α-syn) colocalized with autophagosomal markers p62 and LC3II, but not with lysosomal marker LAMP-1, suggesting that α-syn inclusions are associated with autophagy machinery at the early stages [103]. Moreover, activation or inhibition of lysosomal activity did not alter the levels of P-α-syn, suggesting that matured α-syn aggregates were not efficiently degraded by the lysosomes. These cells also exhibit an impairment in autophagy with defective autophagosome maturation and delivery to the lysosomes. This work suggested that once α-syn aggregates are matured and the pathology is initiated, autophagy is no longer effective and is dysregulated by α-syn [103]. Detrimental effects of α-syn aggregates also occur through interaction of these aggregates with membrane lipids, inducing lysosomal membrane permeabilization and consequent impairment in autophagy [104,105].

In the lysosomal lumen, α-syn alters lysosomal hydrolases activity, such as Cathepsin B, GCase, β-galactosidase, or β-hexosamidase, in midbrain neurons generated from iPSC cell line derived from healthy subjects transfected with WT α-syn or from idiopathic PD patients [77,106]. Strikingly, the levels of total cellular hydrolases were not altered, suggesting a problem in maturation and trafficking of the enzymes rather than a biosynthesis alteration. This idea was confirmed by experiments showing that α-syn modified the location of the key-regulator of ER-Golgi trafficking, Rab1, leading to disruption of the hydrolase trafficking at the Golgi compartment [106]. Interestingly, interaction of α-syn with GCase at its active site in the lysosomal compartment [107], as well as decreased GCase lysosomal activity in presence of WT or A53T α-syn due to alteration in protein maturation [77], further strengthened the risk-association of *GBA* mutations between PD and GD.

Finally, α-syn also dysregulates specific autophagy pathways, such as mitophagy. Choubey and colleagues showed that cortical neurons transfected with mutant A53T α-syn increased levels of healthy mitochondria colocalizing with autophagosomes, and were associated with decreased ATP levels that provoked bioenergetic deficits for the neuron [12]. In this situation, the physiological role of Parkin, which is to mark the mitochondria for their lysosomal addressing, is completely altered [12].

Altogether, these data suggested that the α-syn protein alters the ALP machinery through various deleterious pathways, thus participating in cellular dyshomeostasis and neuronal death, but also exacerbating the pathological synucleinopathy by a dual-loop effect (Figure 1b).

## 4. Role of Autophagy Impairment in Neurodegeneration

Since autophagy plays a role in the accumulation of α-syn, and in turn in the formation of a α-syn aggregates, its contribution in neurodegeneration must be questioned. Autophagic genes, such as *Atg5* and *Atg7*, have been implicated in neurodegeneration. Both *Atg7* or *Atg5* conditionally KO mice displayed neuronal cell death, presence of ubiquitin-positive inclusions, and behavioral deficits [108,109,110], suggesting that autophagy impairment is implicated in neurodegeneration. In synucleinopathies, autophagy dysfunction has been observed through different mechanisms within the cell from the lysosomal stress, to mitophagy defect (a mitochondrial-centered autophagy), and finally in cellular trafficking disturbances that lead to neuronal loss. 

### 4.1. Lysosomal Dysfunction

Neurons require a tight regulation and recycling of proteins involved, for the most part, in neurotransmission, making lysosomal efficacy that much more important for neuronal function and survival [111]. Correct completion of autophagy results in efficient cargo degradation via lysosomal digestion. Issues with lysosomal stress have been shown to affect the elimination of cargo. 

Suitable autophagosomal degradation is based on the fusion of the autophagosomes with the lysosomes to be able to correctly degrade their content. Specific lysosomal proteins have been involved in the formation of the lysosome, such as LAMP-1 and LAMP-2. In LAMP-1 and LAMP-2 double-mutant mice, early embryonic death from E14.5 to E16.5 was observed with accumulation of autophagic vacuoles containing non-degraded material [112]. The same cellular disruptions were observed in fibroblast cell lines derived from these double mutant embryos, but lysosomal enzyme activity and protein degradation rates remained unchanged. These results suggest the important role of both LAMP proteins in lysosome formation that goes beyond structural maintenance of lysosomes.

Lysosomal destruction of the cargo requires a specific acidic environment to activate the hydrolases and proteases to, in turn, digest cargo. Neurotoxic agents such as MPP+ lead to the alkalinization of the lysosomal compartment associated with neurodegeneration [113]. In genetic PD models, such as ATP13A2 KO mice, not only does the loss of this protein cause a lysosomal alkalinization associated with a reduction of lysosomal proteolysis, but also triggers cell death [43]. In addition, when restoring this protein in dopaminergic cell lines, the reestablishment of lysosomal acidification and function leads to a decrease in cell death. Supplementary studies using ATP13A2-null mice were conducted and confirmed that ATP13A2 deficits were enough to induce neurodegeneration. Despite overexpressing α-syn in these mice, no pathology alterations were observed, indicating that ATP13A2 deficits act independently from α-syn [114]. 

Mutations in the LRRK2-encoding gene also indicated an impairment in the lysosomal pH and cathepsin activity [115]. LRRK2 G2019S mutations diminish lysosomal capacities and induce an enlargement of these organelles. These mutations also lead to an increased expression of ATP13A2, suggesting a link between these two lysosomal proteins associated with PD. The pathogenic mechanism originating from LRRK2 mutations remains unclear in PD pathogenesis. As LRRK2 KO mice present a little neurodegeneration in the brain [116], double knockouts of both LRRK proteins (LRRK1 and LRRK2) have been generated to elucidate the potential compensation of these proteins [117]. These double-mutant mice showed early mortality as well as dopaminergic cell loss accompanied by autophagy dysfunction, highlighting the importance of this protein in the regulation of ALP and neurodegeneration. 

In line with acidification of the lysosomes, deficiencies in lysosomal proteases and hydrolases have been observed in PD models. First, mutations in the lysosomal protease CathD in *C. elegans*, as well as in mice, were shown to have a deleterious impact on autophagy and to cause the formation of α-syn aggregates and neurodegeneration in the CNS [118]. In addition, mutations in the *GBA1* gene have demonstrated parkinsonian symptoms linked to neurodegeneration [119]. In the case of MSA pathology, after differentiation of iPSC-derived dopaminergic neurons from MSA patients, impairment of autophagy and particularly of lysosomal enzymes were also observed [50].

The MPTP mouse model recapitulates some of the major features of PD, such as modifications in mitochondrial structure and function, reactive oxygen species (ROS) production, and activation of apoptotic pathways, which induce dopaminergic cell death. In this model, a pathogenic lysosomal depletion occurs prior to an autophagosomal accumulation contributing to PD-related dopaminergic neurodegeneration [38]. 

Taken together, these studies emphasized the occurrence of lysosomal defects in in vitro and in vivo PD models as a key player in the initiation of neurodegeneration (Figure 2a).

### 4.2. Mitophagy Defects

Main cellular functions of the cell also reside in mitochondria, as they are the key organelles in the cell for the production of energy [120]. Defects in mitochondrial functions have been associated with genetic forms of synucleinopathies, particularly in PD with protein deficiencies, such as PINK1 and Parkin [121,122,123,124]. In the cell, degradation of mitochondria via the autophagic pathway, known as mitophagy, is essential for the elimination of defective mitochondria. When mitochondria are defective, there is recruitment of PINK1 to the outer membrane, where it accumulates and in turn phosphorylates Parkin, leading to its activation. The activation of Parkin induces the recruitment of a polyubiquitin chain. This polyubiquitin chain is then phosphorylated by PINK1, leading to the recruitment of autophagy receptors, activating the induction of autophagy and the clearance of the defective mitochondria [125].

In PD, this mitochondrial clearance does not function properly. Studies using deletions in genes of Parkin and PINK1 were first described in *Drosophila melanogaster* [121,122,123]. Given the central role of these two proteins in mitophagy, it is important to understand the contribution of this specific autophagic pathway in the induction of neurodegeneration. In Parkin-deleted adult mice, Stevens and colleagues observed declines in mitochondrial mass as well as age-dependent loss of dopaminergic neurons [126]. Recently, Sliter and collaborators showed that Parkin and PINK1 defects induced an inflammatory response that can play a role in neurodegeneration [127]. Nevertheless, the exact contribution of Parkin or PINK1 defective mitophagy in neurodegeneration need further exploration [128]. DJ-1, a PD-linked protein associated with familial cases, seems to also play a role in mitophagy, as it is able to rescue the phenotype of PINK1 mutants in Drosophila, but not in Parkin mutants. This suggests that DJ-1 could act between PINK1 and Parkin [129]. Interestingly, DJ-1 overexpression increased α-syn clearance and decreased neuronal death [130]. In DJ-1-deficient cells, increased levels of oxidative stress via glutathione inhibition were responsible for an increased sensitivity of cells to ROS, acting on the correct functions of the cell and its viability [131]. Loss of DJ-1 has also been shown to disrupt CMA, in addition to inducing mitochondrial defects [132]. Finally, Li and colleagues observed that *GBA* heterozygous mutations impair mitophagy by altering the mitochondrial priming to the lysosomes, indicating that *GBA* mutations not only affect ALP, but also mitochondrial recycling [133].

Thus, impairment of the mitophagy process seems to contribute to oxidative stress conditions, ultimately leading to neurodegeneration (Figure 2b).

### 4.3. Vesicular Trafficking Defects

Finally, trafficking defects have been observed in α-syn-related pathologies. Trafficking of endosomes and autophagosomes is essential in the autophagic pathway as it allows the transfer of the cargo to the ALP. VPS35 is implicated in endosomal trafficking. Mutations in VPS35 have been involved in late-onset PD [134]. In VPS35 PD-causing mutations, localization of the WASH complex within the endosomes was reduced and autophagy defects associated with cell death occurred [135]. WASH complex is involved in binding with retromers and inducing the formation of F-actin patches to guide cargo for transport. This complex has been shown to be involved in the induction of autophagy, particularly in the autophagosome formation step [135]. The VPS35 protein seemed to be essential in the regulation of the WASH complex to maintain cargo trafficking. Parkin has also emerged as a key player in endosomal trafficking. Williams and colleagues showed a novel interaction between Parkin and VPS35, in which Parkin mediated the ubiquitination of VPS35 [136]. VPS35 ubiquitination acts directly on WASH-dependent cargo sorting and alters endosomal trafficking, thus affecting correct autophagic function, and finally playing a role in neurodegeneration.

The interaction between the two proteins Beclin-1 and Vps34, implicated in cargo trafficking, is the launching point to initiate autophagy. Huang and colleagues showed that HMGB1 protein played an indirect role on autophagy impairment via the malformation of the Beclin1-Vps34 complex in rotenone-treated cells, which inhibits autophagosome formation [137]. HMGB1 overexpression and rotenone exposure had the same effects on the cells—they induce cellular shrinkage and decrease cell viability.

As membrane components, lipid and cholesterol metabolism have also been shown to be players in vesicle integrity and trafficking. Cholesterol has been implicated specifically in autophagosome trafficking. Accumulation of cholesterol observed in *GBA* N370S mutant fibroblasts leads to an accumulation of multilamellar bodies and a disruption of autophagy [138], increasing their susceptibility to apoptosis. In *GBA1*-deficient cells, an impairment of secretory autophagy can also be seen [139]. Lysosomal membrane ceramide species are produced by the GCase enzyme and then are converted into other lipids, such as fatty acids, by acid ceramidase [140]. Kim and colleagues overexpressed acid ceramidase in WT cells, which induced the accumulation of autophagic markers (p62, LC3, Rab7), as well as cell death [139]. Inhibiting acid ceramidase in *GBA1*-deficient cells led to a partial rescue of the lipid abnormalities, as well as decreased neurodegenerative markers [139]. 

Trafficking defects compromise autophagosomes delivery to the lysosomal compartment, thus inhibiting correct protein turnover, finally leading to neurodegenerative processes (Figure 2c).

## 5. Autophagy Recovery as a Therapeutic Target

We have reported here that ALP impairment plays a pivotal role in α-syn accumulation and release, as well as in neurodegenerative processes. Many studies have highlighted the importance of increasing autophagy activity or restoring its functionality as a therapeutic strategy in the field of synucleinopathies [141].

### 5.1. mTOR-Dependent Targeting

ALP is regulated upstream by the mTOR protein, in which the inhibition activates autophagy. Molecules such as rapamycin increase mTOR inhibition, and thus increase the autophagy levels within the cell. This molecule has been tested in multiple models of PD, including, but not limited to, rotenone-exposed cells and MPP+-treated cells, MPTP mouse model, and α-syn transgenic rodents [40,142,143,144]. All these models have shown beneficial effects of rapamycin administration in both neurotoxin-induced PD models and α-syn overexpressing models. Specifically, there has been shown to be a reduction in dopaminergic cell death, decreased levels of phosphorylated α-syn, and reduced mitochondrial dysfunction. To target the mTOR pathway directly, some studies have also targeted the downstream mTOR target, transcription factor EB (TFEB). The mTOR inhibition activates the transport of TFEB to the nucleus, in which it binds DNA to regulate ALP gene expression [145]. Attempts at activating TFEB have been done through molecules such as FDA-approved 2-hydropropyl-ß-cyclodextrin, which are capable of increasing autophagy levels, as well as α-syn clearance in neuroglioma cell lines [146]. Gene therapy using adeno-associated viral mediated TFEB overexpression in α-syn-overexpressing rat conferred a protective effect on rat midbrain neurons, associated with an increased clearance of pathologic α-syn through autophagic recovery [147]. Natural compounds such as pomegranate extracts enhanced TFEB activity, reversing the effects of α-syn mediated neurodegeneration in neuroblastoma cells through induction of both mitophagy and autophagy [148].

Neuroinflammation also contributes to neurodegeneration in synucleinopathies, but its role in triggering autophagy defects has not yet been completely elucidated. Nonetheless, innate immunity receptors, such as Toll-like receptors (TLR), have been suggested to play a role in neurodegeneration, as well as neuroinflammation, through autophagic dysregulation dependent on the mTOR pathway [149]. In fact, TLR2 expression has been shown to be increased in PD and LBD patients, as well as models of these diseases [150]. Activating TLR2 expression in vitro induced an accumulation of α-syn aggregates and increased neurotoxicity through autophagic dysregulation. On the other hand, knockdown of TLR2 in a mouse model of PD allowed an improvement in α-syn-induced pathology and alleviated motor deficits, suggesting that TLR2 can represent a novel interesting therapeutic target.

### 5.2. mTOR-Independent Targeting

Other molecules aimed to target mTOR-independent pathways through various mechanisms. A pioneer molecule for this strategy is the invertebrate disaccharide trehalose, which targets SLC2A transporters and induces an AMPK-dependent increase of autophagy. Trehalose has been tested in multiple models of PD, ranging from drug-induced pathology to α-syn overexpressing in vitro and in vivo models [100,151,152]. Using this molecule, there has been shown to be an increase in α-syn clearance, as well as a reduction in cell loss and neuroinflammation. Chronic treatments of natural caffeine molecules on α-syn fibrils-injected mice decreased α-syn positive inclusions, cellular apoptosis, microglial activation, and neuronal loss, all mediated by an increase in autophagy [153]. Lithium constitutes another strategy to enhance autophagy in an mTOR-independent pathway in different models via the inhibition of inositol phosphatase [154,155]. Lithium treatment was shown to increase the autophagy marker LC3 and to decrease the dopaminergic cell loss in the SN of MPTP mice [155]. Additionally, Hou and colleagues observed mitochondrial protection, in addition to autophagy recovery, increasing cell viability after treatment of rotenone-exposed SH-SY5Y neuroblastoma cells with lithium [154]. Nonetheless, lithium is a molecule that is primarily used to treat bipolar disorders, indicating that possible unwanted side effects could emerge from the use of this molecule. Other molecules, such as calcium channel blockers, potassic ATP channel blockers (Minoxidil), and Gi signaling activators (Clonidine), have also been shown to be effective in inducing autophagy and clearing protein aggregates through regulation of cAMP levels [156]. Finally, inhibition of the autophagy blocker c-Abl using Nilotinib has been tested in PD patients who showed an increased clearance of α-syn in biological fluids associated with reduced symptomatology during clinical trials [157,158].

To target lysosomal activity directly, the use of acidic nanoparticles has been tested for their potential to restore lysosomal pH [113,159,160]. Bourdenx and colleagues tested the effect of nanoparticles in MPP+-treated cells, ATP13A2 mutant fibroblasts, and MPTP-treated mice. In all the mentioned models, the nanoparticles had the capacity to restore lysosomal pH, lysosomal function, and finally inhibit dopaminergic cell death in mice brains [113]. To mediate the lysosomal disruption caused by a loss-of-function of ATP13A2, strategies involving metal homeostasis have been used as therapeutic strategies. Tsunemi and coworkers recently observed that loss of ATP13A2 function in fibroblasts induced a Ca^2+^ dyshomeostasis in the lysosomes, in turn leading to exosomal disruptions and autophagic impairment [61]. TRPML1, the principal receptor responsible for Ca^2+^ import in lysosomes, was used as a target to modulate Ca^2+^ levels and restore correct function of this organelle. Other studies have also used strategies to target metals such as zinc, as well as cAMP to restore lysosomal acidification [161].

Knowing the importance of autophagy-related genetic mutations and associated protein loss-of-function in PD pathology, restoration of protein activity implicated in synucleinopathies have also demonstrated encouraging beneficial effects. Targeting GCase activity using ambroxol in *GBA1* mutant fibroblasts [162,163] restored lysosomal function and reduced oxidative stress. Preclinical trials using ambroxol are currently being conducted on PD dementia patients to observe the possible beneficial effects of this molecule on GCase activity and PD pathology [164]. Given the implication of Beclin-1 in inducing autophagy, targeting this protein to increase autophagic clearance has been tested in different models. Using activators of Beclin-1 in multiple PD models has been shown to increase α-syn clearance, decreasing cellular toxicity [65,165,166]. Despite the interaction of Beclin-1 with proteins able to induce apoptosis, such as Bcl2, the activation or overexpression of Beclin-1 has not seemed to increase the amount of cell death. Possibilities of targeting CMA are more limiting than targeting macroautophagy. Nonetheless, downstream factors of the CMA pathway could be an appealing alternative to macroautophagy. CMA targeting has principally been through the lysosomal marker LAMP-2A and its chaperone Hsc70. Overexpression of LAMP-2A has been induced in cellular models of PD using SH-SY5Y cells, in primary neuron cultures, as well as in rats [167]. In these models, an increase in the expression of the lysosomal protein LAMP-2A increased CMA activity, thus decreasing α-syn accumulation, allowing neuroprotection [167]. In addition to overexpressing LAMP-2A, retinoic acid receptors have been shown to be inhibitors of CMA. This inhibition decreased oxidative stress and protein toxicity, proving the interest of increasing the CMA pathway [168]. Mitochondrial oxidative stress also plays an important role in autophagy impairment in synucleinopathies. The antioxidant properties of the DJ-1 protein have been previously reported, as it could decrease oxidative stress and restore autophagy [169,170,171]. Increasing DJ-1 expression in astrocytes co-cultured with neurons proved to have beneficial effects against rotenone-induced cell oxidation [169]. This was also shown in vivo using a rotenone-treated rat model, where DJ-1 was specifically increased in astrocytes [172]. This DJ-1 overexpression allowed an inhibition of dopaminergic cell loss, a decrease in neuroinflammation, as well as an inhibition of CMA deficits. These studies indicate DJ-1 as an interesting genetic target because it acts at multiple levels of synucleinopathies: α-syn interaction, mitophagy activation, and oxidative stress reduction.

Pupyshev and colleagues, in an attempt to test the synergistic effect of mTOR-dependent and -independent targeting, combined the use of trehalose and rapamycin in a MPTP mouse model of PD [173]. By combining both molecules, this study showed an even more pronounced improvement in both the number of protected dopaminergic neurons and motor function recovery.

Knowing the crucial role that autophagy plays in the maintenance of cellular homeostasis, reestablishing its proper function in synucleinopathies was proven to be efficient in clearance of pathological α-syn aggregates and neuroprotection. However, this strategy could be detrimental to the well-being of neurons, as increasing the ALP could also target other proteins’ elimination. Nevertheless, autophagy remains a promising and relevant target that could be at the source of possible therapies for synucleinopathies, as well as other protein accumulation-based neurodegenerative diseases.

## 6. Conclusions

Here, we summarized distinct pieces of evidence suggesting that autophagy is involved in synucleinopathy. The presence of autophagy alterations was first provided by genetic and post-mortem studies on brains of PD and MSA patients. Since then, experimental studies attempted to demonstrate the role of autophagy in the pathology of synucleinopathy. First, α-syn is mainly degraded by both macroautophagy and chaperone-mediated autophagy. Thus, autophagy defects induce intracellular α-syn accumulation, participating in its aggregative state towards the formation of α-syn-positive intracytoplasmic inclusions. In addition to this aggregative-prone phenomenon, autophagy defects also increase the α-syn secretion by the non-autophagic exosomal pathway, leading to increased cell-to-cell transmission of the protein, and thus the propagation of the α-syn-linked pathology in different brain regions of the CNS. On the other hand, autophagy defects also cause detriment effects in cellular homeostasis: (i) lysosomal impairment through structural or functional defects leads to accumulation of non-degraded products and increased production of ROS; (ii) decreased mitophagy leads to neuronal bioenergetic imbalance; and (iii) defective cargo trafficking impairs the addressing of vesicles to lysosomal clearance. All these detrimental cellular conditions lead to neurodegeneration. Finally, increased evidence demonstrated that inducing the autophagy pathways by natural or chemical compounds, as well as genetic approaches, has become a relevant therapeutic approach to counteract the deleterious effects of autophagy impairment in synucleinopathy.

## Figures and Tables

**Figure 1 cells-08-00565-f001:**
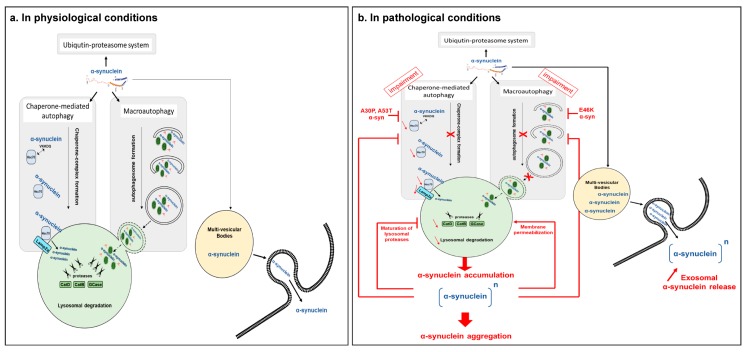
Dual loop between autophagy lysosomal pathway and α-synuclein. (**a**) In physiological conditions, autophagy machinery participates in α-synuclein degradation through both macroautophagy and chaperone-mediated autophagy. The intracellular α-synuclein levels are maintained at basal conditions and exosomal α-synuclein release is low. (**b**) In pathological conditions, autophagy impairment leads to decreased α-synuclein clearance. Intracellular α-synuclein accumulates in the cells and participates in the α-synuclein aggregative process. To compensate for this defect, the cell attempts to eliminate α-synuclein through the exosomal pathway, inducing increased released α-synuclein amounts and cell-to-cell propagation of the protein. Reciprocally, accumulated pathological α-synuclein induces cell toxicity mediated in part by autophagy impairment, diving the cell into an endless cycle of dyshomeostasis. Hsc70: cytosolic chaperone heat-shock cognate 70kDa protein; CatD: Cathepsin D; CatB: Cathepsin B; GCase: β-glucocerebrosidase.

**Figure 2 cells-08-00565-f002:**
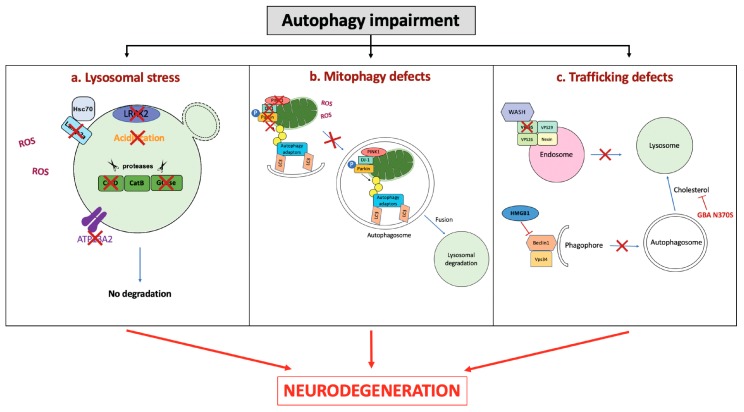
Autophagy impairment induces neurodegeneration. Autophagy can be impaired at different levels, namely: (**a**) Lysosomal stress; (**b**) mitophagy defects; (**c**) vesicular trafficking defects. (**a**) Lysosomal stress, characterized by either lysosomal alkalinization, lysosomal structural modifications, or defective lysosomal proteases, blocks the correct degradation of the cargo. (**b**) Mitophagy relies on actors such as PINK1, DJ-1, and Parkin, which allow a correct priming of defective mitochondria to be addressed to autophagy. Alterations in these proteins induce decreased mitophagy. (**c**) Vesicular trafficking is the starting point for the autophagosome formation though the recruitment of the WASH complex. Mutations in VPS35 inhibits the WASH complex, and thus autophagosome formation. HMGB1 overexpression inhibits the Beclin1-Vps34 binding, and thus the phagophore formation. Mutations in *GBA* genes can also modify cholesterol, impairing autophagosome trafficking. GCase: β-glucocerebrosidase; CatD: Cathepsin D; ROS: reactive oxygen species.

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
