# Peer review of "Autophagy in Synucleinopathy: The Overwhelmed and Defective Machinery"

_cells, 2019, doi:10.3390/cells8060565_

Round 1

Reviewer 1 Report

This is a comprehensive review on the topic of autophagy and neurodegeneration. Specifically, the authors focus on synucleinopathy, which is often associated with autophagy defects. The authors discussed clinical evidences supporting the role of autophagy in synucleinopathy through genetic, post-mortem analyses. The authors discuss both macroautophagy and chaperone-mediated autophagy, and how their defects can lead to alpha-synuclein accumulation, aggregation, propagation and subsequent induction of neurodegenerative pathologies. The authors also discuss about the therapeutic strategies.

Specific comments:

3.1. The authors discuss about both accumulation and aggregation. Subtitle should be updated.

4.1. Lysosomal stress: lysosomal dysfunction could be more appropriate subtitle for this subsection.

4.2. Mitophagy defect section -- It is well written that PINK/Parkin/DJ1 are important for mitophagy. It would be even better if the authors describe about how these mutations can cause neurodegeneration. Is there clear evidence that ROS is the major target of mitophagy suppressing neurodegeneration? 

5.2 mTOR-independent targeting -- In addition to the drugs that the authors describe, calcium channel blockers, K+ATP channel modulators, G(i) signaling modulators were also effective in modulating autophagic flux (PMID: 18391949). These can be incorporated in this section.

Author Response

Point-by-point answer to Reviewer #1

This is a comprehensive review on the topic of autophagy and neurodegeneration. Specifically, the authors focus on synucleinopathy, which is often associated with autophagy defects. The authors discussed clinical evidences supporting the role of autophagy in synucleinopathy through genetic, post-mortem analyses. The authors discuss both macroautophagy and chaperone-mediated autophagy, and how their defects can lead to alpha-synuclein accumulation, aggregation, propagation and subsequent induction of neurodegenerative pathologies. The authors also discuss about the therapeutic strategies.

Specific comments:

3.1. The authors discuss about both accumulation and aggregation. Subtitle should be updated.

The subtitle 3.1 has been modified as suggested: â€ś3.1. ALP processes participate in α-syn accumulation and aggregation”(line 186).

4.1. Lysosomal stress: lysosomal dysfunction could be more appropriate subtitle for this subsection.

The subtitle 4.1 has been modified as suggested: â€ś4.1. Lysosomal dysfunction”(line 384).

4.2. Mitophagy defect section -- It is well written that PINK/Parkin/DJ1 are important for mitophagy. It would be even better if the authors describe about how these mutations can cause neurodegeneration. Is there clear evidence that ROS is the major target of mitophagy suppressing neurodegeneration?

We took into consideration the comment of the reviewer and added few sentences about the current knowledge on the mechanism of action by which these mutations can cause neurodegeneration (lines 447-450). One study suggested that inflammatory response may be involved but little is known about the mechanism of action linking PINK1/Parkin mutations to neurodegeneration.

5.2 mTOR-independent targeting -- In addition to the drugs that the authors describe, calcium channel blockers, K+ATP channel modulators, G(i) signaling modulators were also effective in modulating autophagic flux (PMID: 18391949). These can be incorporated in this section.

We have added a sentence associated with this reference on mTOR-independent autophagy inducers leading to efficient clearance of Huntingtin protein in Huntington’s Disease (lines 558-560, reference number 156).

Reviewer 2 Report

Review of the manuscript “Autophagy in synucleinopathy: the overwhelmed and defective machinery” by Marie-Laure Arotcarena and coauthors submitted to Cells, MDPI. 

Parkinson’s disease is the second most frequent neurodegenerative disorder, which is associated with the accumulation of synuclein in neuronal tissues. There is no efficient treatment of this disease affecting the course of the pathology. The authors discuss a concept stating that the most important contribution to this disorder is the impairment of autophagy which is primarily responsible for the alpha-synuclein accumulation and propagation. This field is very important and the data discussed in the manuscript will be interesting for “Cells” readership.  The manuscript is well written and contains new ideas. The following corrections should be done:

Abstract (lines 13-14) and Introduction (Lines 37-38, lines 42-43): Speaking about synucleinopathies the authors write about Parkinson’s disease and Multiple System Atrophy. They should mention, at last briefly, dementia with Lewy bodies (DLB).

Abstract, line 17: “Here, we aimed to report evidence that…”  This statement looks like the authors want to present new experimental evidence. However their manuscript is a review. So it will be more appropriate to replace “to report evidence” on “to discuss” or anything more suitable for a review article.

Keywords: the authors should add the following keywords: Parkinson’s disease, synucleinopathies

Introduction. Lines 32-33. After the sentence “The major neuropathological hallmark of PD is the presence of α-syn-positive intracytoplasmic inclusions called Lewy Bodies (LB) which accumulate in neurons [2]” the authors should add a reference on a recent review on Parkinson’s disease: Emamzadeh  et al., “Parkinson’s disease: Biomarkers, Treatment, and Risk Factors.” Front. Neurosci., 30 August 2018 | https://doi.org/10.3389/fnins.2018.00612

Introduction, line 47. Why the authors do not mention here mutations in alpha synuclein which increase the propensity to aggregation?

Introduction, lines 47-49: α-Syn 47 degradation is ensured by both the ubiquitin-proteasome system [A] and by the autophagy-lysosomal pathways (ALP) protein degradation machinery [B]

The authors should add the following references to this sentence:

[A]. beta-Synuclein prevents proteasomal inhibition by α-synuclein but not γ-synuclein. J Biol Chem, 2005. 4; 280:7562-9.

[B] Rivero-RĂ­os et al., Targeting the Autophagy/Lysosomal Degradation Pathway in Parkinson's Disease. Curr Neuropharmacol. 2016;14(3):238-49.

Introduction, line 62 “Genetic studies have been conducted for PD as ~10% of PD cases are familial forms with hereditary parkinsonism and earlier onset of the disease”. This is an awkward sentence, which should be rewritten.  

Introduction, line 62. 2.1. In the chapter “Genetic evidence of autophagic involvement in familial PD cases” the authors do not mention a contribution of mutant forms of a-synuclein in autophagy, described, for example in the following articles:

-Yan et al., E46K Mutant α-Synuclein Is Degraded by Both Proteasome and Macroautophagy Pathway. Molecules. 2018 Nov 1;23(11).

-Lei et al., A30P mutant α-synuclein impairs autophagic flux by inactivating JNK signaling to enhance ZKSCAN3 activity in midbrain dopaminergic neurons. Cell Death Dis. 2019 Feb 12;10(2):133. doi: 10.1038/s41419-019-1364-0.

Figure 1: The authors should increase the text on the figure, especially on Fig. 1b. What is above “Exosomal a-synuclein release”? It is impossible to see. The same concerns a fragment above “a-synuclein aggregation”. The writing in the vesicles are not readable.

Conclusion, Lines 605 – 607 :”autophagy defects also bring detrimental cellular homeostasis with lysosomal stress, production of ROS, excessive mitophagy, defective cargo trafficking, in turn leading to cell death and neurodegeneration”.

This sentence refers to too many players, while their association and role in disease is not explained. The statement should be clarified in details, explaining connections between different processes.

Author Response

Point-by-point answer to Reviewer #2

Parkinson’s disease is the second most frequent neurodegenerative disorder, which is associated with the accumulation of synuclein in neuronal tissues. There is no efficient treatment of this disease affecting the course of the pathology. The authors discuss a concept stating that the most important contribution to this disorder is the impairment of autophagy which is primarily responsible for the alpha-synuclein accumulation and propagation. This field is very important and the data discussed in the manuscript will be interesting for “Cells” readership.  The manuscript is well written and contains new ideas. The following corrections should be done:

Abstract (lines 13-14) and Introduction (Lines 37-38, lines 42-43): Speaking about synucleinopathies the authors write about Parkinson’s disease and Multiple System Atrophy. They should mention, at last briefly, dementia with Lewy bodies (DLB).

We have now mentioned dementia with Lewy bodies in the abstract (line 13) and in the introduction (line 28 and lines 34-37).

Abstract, line 17: “Here, we aimed to report evidence that…”  This statement looks like the authors want to present new experimental evidence. However, their manuscript is a review. So, it will be more appropriate to replace “to report evidence” on “to discuss” or anything more suitable for a review article.

We have now replaced the term “report evidence” with the term “discuss” (line 17) in the revised abstract.

Keywords: the authors should add the following keywords: Parkinson’s disease, synucleinopathies

We have now added the three following keywords: Parkinson’s disease, Multiple system atrophy and synucleinopathy (line 22).

Introduction. Lines 32-33. After the sentence “The major neuropathological hallmark of PD is the presence of Î±-syn-positive intracytoplasmic inclusions called Lewy Bodies (LB) which accumulate in neurons [2]” the authors should add a reference on a recent review on Parkinson’s disease: Emamzadeh et al., “Parkinson’s disease: Biomarkers, Treatment, and Risk Factors.” Front. Neurosci., 30 August 2018 | https://doi.org/10.3389/fnins.2018.00612

We have now added this reference (number 3) at the end of the sentence (line 34).

Introduction, line 47. Why the authors do not mention here mutations in alpha synuclein which increase the propensity to aggregation?

We have now mentioned the mutations in the sentence “(ii) post-translational modifications or mutations favoring α-syn aggregation”in the introduction (line 50).

Introduction, lines 47-49: Î±-Syn 47 degradation is ensured by both the ubiquitin-proteasome system [A] and by the autophagy-lysosomal pathways (ALP) protein degradation machinery [B]

The authors should add the following references to this sentence:

[A]. beta-Synuclein prevents proteasomal inhibition by Î±-synuclein but not Îł-synuclein. J Biol Chem, 2005. 4; 280:7562-9.

[B] Rivero-RĂ­os et al., Targeting the Autophagy/Lysosomal Degradation Pathway in Parkinson's Disease. Curr Neuropharmacol. 2016;14(3):238-49.

We have now added these two references (4 and 5) as proposed for this sentence (lines 51-52) in the introduction.

Introduction, line 62 â€śGenetic studies have been conducted for PD as ~10% of PD cases are familial forms with hereditary parkinsonism and earlier onset of the disease”. This is an awkward sentence, which should be rewritten.  

We have now replaced this sentence: â€śGenetic studies have been conducted for PD identifying familial forms of the disease representing ~10% of the PD cases with hereditary parkinsonism and earlier onset of the disease”(lines 67-68).

Introduction, line 62. 2.1. In the chapter “Genetic evidence of autophagic involvement in familial PD cases” the authors do not mention a contribution of mutant forms of a-synuclein in autophagy, described, for example in the following articles:

-Yan et al., E46K Mutant Î±-Synuclein Is Degraded by Both Proteasome and Macroautophagy Pathway. Molecules. 2018 Nov 1;23(11).

-Lei et al., A30P mutant Î±-synuclein impairs autophagic flux by inactivating JNK signaling to enhance ZKSCAN3 activity in midbrain dopaminergic neurons. Cell Death Dis. 2019 Feb 12;10(2):133. doi: 10.1038/s41419-019-1364-0.

We have now updated this chapter mentioning the contribution of mutant forms of a-synuclein (A53T, A30P and E46K) in autophagy with associated references (lines 71 – 77; References 10 and 11). One sentence has been also added in the part 3.3 regarding the A30P mutant a-synuclein effect on autophagy (lines 330-331).

Figure 1: The authors should increase the text on the figure, especially on Fig. 1b. What is above “Exosomal a-synuclein release”? It is impossible to see. The same concerns a fragment above “a-synuclein aggregation”. The writing in the vesicles are not readable.

We have now modified the Figure 1, replacing the unreadable sequence of a-synuclein by the word a-synuclein in blue as followed: [α-synuclein]n.

Conclusion, Lines 605 – 607:” autophagy defects also bring detrimental cellular homeostasis with lysosomal stress, production of ROS, excessive mitophagy, defective cargo trafficking, in turn leading to cell death and neurodegeneration”.

This sentence refers to too many players, while their association and role in disease is not explained. The statement should be clarified in details, explaining connections between different processes.

We have now updated the conclusion by taking into consideration the comment of the reviewer (lines 624-629).